# Potential of Phototrophic Purple Nonsulfur Bacteria to Fix Nitrogen in Rice Fields

**DOI:** 10.3390/microorganisms10010028

**Published:** 2021-12-24

**Authors:** Isamu Maeda

**Affiliations:** Department of Applied Biological Chemistry, School of Agriculture, Utsunomiya University, 350 Minemachi, Utsunomiya 321-8505, Japan; i-maeda@cc.utsunomiya-u.ac.jp; Tel.: +81-28-649-5477

**Keywords:** nitrogen fixation, nitrogenase, purple nonsulfur bacteria, anoxygenic phototrophic bacteria, paddy, rice, biofertilizer, *Rhodopseudomonas* spp., *Rhodobacter* spp., *Rhodospirillum* spp.

## Abstract

Biological nitrogen fixation catalyzed by Mo-nitrogenase of symbiotic diazotrophs has attracted interest because its potential to supply plant-available nitrogen offers an alternative way of using chemical fertilizers for sustainable agriculture. Phototrophic purple nonsulfur bacteria (PNSB) diazotrophically grow under light anaerobic conditions and can be isolated from photic and microaerobic zones of rice fields. Therefore, PNSB as asymbiotic diazotrophs contribute to nitrogen fixation in rice fields. An attempt to measure nitrogen in the oxidized surface layer of paddy soil estimates that approximately 6–8 kg N/ha/year might be accumulated by phototrophic microorganisms. Species of PNSB possess one of or both alternative nitrogenases, V-nitrogenase and Fe-nitrogenase, which are found in asymbiotic diazotrophs, in addition to Mo-nitrogenase. The regulatory networks control nitrogenase activity in response to ammonium, molecular oxygen, and light irradiation. Laboratory and field studies have revealed effectiveness of PNSB inoculation to rice cultures on increases of nitrogen gain, plant growth, and/or grain yield. In this review, properties of the nitrogenase isozymes and regulation of nitrogenase activities in PNSB are described, and research challenges and potential of PNSB inoculation to rice cultures are discussed from a viewpoint of their applications as nitrogen biofertilizer.

## 1. Introduction

Global energy demand has increased partly due to the rise of fertilizer consumption and the shift toward more energy-intensive fertilizers [1]. As biosynthesis of ammonia in nitrogen fixation proceeds using nitrogen gas under natural environmental conditions [2], use of diazotrophic bacteria as biofertilizer would reduce chemical fertilizer consumption and contribute to sustainable agriculture. Representatives of such diazotrophic bacteria are root nodule bacteria, which provide the nitrogen source to host legume [3]. Carbon sources such as malic acid, fumaric acid, and succinic acid are supplied through plant glycolysis to the nodule bacteroids, which utilize these organic acids for the energy to fuel nitrogenase activity [4]. Besides the symbiotic nitrogen fixation in legume root–nodule symbiosis, it has been estimated in the surface of rice paddy soil that approximately 6–8 kg N/ha/year might be accumulated as a result of phototrophic nitrogen fixation [5]. The green and purple bacteria, the heliobacteria, many cyanobacteria, and the unusual chlorophyll-containing rhizobia are phototrophic diazotrophs [6]. Purple sulfur and purple nonsulfur bacteria (PNSB) have nitrogen-fixing ability in the light and can grow using nitrogen gas as the sole nitrogen source [7,8]. In these bacterial groups, activity of nitrogenase that catalyzes nitrogen fixation is highly regulated by light, oxygen, and ammonia [9,10]. Purple sulfur bacteria are photoautotrophs, but they are poorly equipped for metabolism and growth in the dark [11]. By contrast, because of diverse capacities for dark metabolism and growth, nitrogen fixation in PNSB occurs both under light anaerobic and dark microaerobic conditions [8,12]. PNSB have been often isolated from paddy soil, rice field soil, and freshwater ponds [13], indicating possibility that they might play an important role on nitrogen cycling in these habitats. This review describes the current knowledge regarding nitrogenase isozymes and regulation of nitrogenase activities in PNSB and focuses on achievements and future directions in their applications as biofertilizer toward the establishment of sustainable agriculture.

## 2. Nitrogenase Isozymes

Nitrogenase comprises two component proteins usually referred to as Fe protein (dinitrogenase reductase) and MoFe protein (dinitrogenase) [14,15]. These designations originate from the metal compositions of the respective component proteins of the conventional Mo-nitrogenase. Mo-nitrogenase is found in all diazotrophs and is the only nitrogenase reported in diazotrophs that forms nitrogen-fixing symbioses with higher plants [16]. Fe protein is encoded by *nifH*, whereas MoFe protein has two different subunits encoded by *nifD* and *nifK* [17]. In addition to Mo-nitrogenase, two alternative nitrogenases, V-nitrogenase and Fe-nitrogenase, are known to contribute to asymbiotic nitrogen fixation by soil bacteria such as *Azotobacter vinelandii* [18,19,20]. The genes designated for V-nitrogenase are *vnfH*, *vnfD*, *vnfG*, and *vnfK*, and the genes designated for Fe-nitrogenase are *anfH*, *anfD*, *anfG*, and *anfK*, where *vnfG* and *anfG* encode additional subunits. VnfG is required for processing apodinitrogenase to functional dinitrogenase [21]. It has been shown that a significant fraction of nitrogen fixation by free-living soil diazotrophs such as a PNSB *Rhodopseudomonas palustris* is contributed by the alternative nitrogenases in addition to Mo-nitrogenase [16,22]. As shown in *R. palustris* CGA009 that synthesizes Mo-, V-, and Fe-nitrogenases [23], alternative nitrogenases have been reported in phototrophic bacteria including PNSB, free-living cyanobacteria [24,25], and symbiotic cyanobacteria [26], but have not been found in purple sulfur bacteria in VnfH and AnfH homologue searches using BLASTp (https://blast.ncbi.nlm.nih.gov/Blast.cgi, accessed on 22 October 2021). Interestingly, it has been reported that no alternative nitrogenases are detectable, and nitrogen fixation is exclusively catalyzed by Mo-nitrogenase even at low molybdenum conditions in situ, where purple sulfur bacteria are responsible for high nitrogen fixation rates [27].

### 2.1. Fe-Nitrogenases in PNSB

Existence of Fe-nitrogenase in *Rhodobacter capsulatus* has been demonstrated with diazotrophic growth of a *nifHDK* deletion mutant in media containing <0.05 ppb Mo [28]. The biochemical characterization studies using the purified enzyme have revealed its lower stability and specific activity with nitrogen and acetylene as substrates and higher ratio of ethane/ethylene formation upon acetylene reduction compared to those of Mo-nitrogenase [29], as well as the activity ratio of 7.5 mol H_2_ produced/mol N_2_ reduced, which is higher than 1.0 mol H_2_ produced/mol N_2_ reduced for Mo-nitrogenase [30]. Fe-nitrogenase has been also purified and characterized from a *nifH* mutant of *Rhodospirillum rubrum* [31]. The enzyme complex reduces protons better than nitrogen and reduces acetylene to a mixture of ethylene and ethane. The gene clusters related to Fe-nitrogenase in addition to V-nitrogenase have been shown in *R. palustris* CGA009 [32]. A Δ*nifH* Δ*vnfH* strain had the phenotype of an Fe-nitrogenase-expressing strain. It had low rates of acetylene reduction and produced ethane and ethylene at an ethane/ethylene ratio of about 0.055. Wild-type Fe-nitrogenase in *R. palustris* catalyzes reduction of carbon dioxide to form methane, simultaneously with reduction of nitrogen gas and protons [33]. It has been suggested that Fe-nitrogenase might support growth of microbial communities in nature by producing significant quantities of methane [34].

### 2.2. V-Nitrogenases in PNSB

Analyses using gene-deletion mutants in *R. palustris* indicates that the addition of exogenous V (10 μM) allows the Δ*nifH nifD*::Tn5 Δ*anfA* strain to grow diazotrophically and to reduce acetylene to ethane and ethylene at ratios of about 0.015 [32]. Ethane formation during the acetylene reduction is indicative of active V- and Fe-nitrogenases. The isotopic acetylene reduction assay, which can specifically measure ethylene yield in nitrogenase-catalyzed acetylene reduction, has demonstrated measurements of V- and Fe-nitrogenase activities in *R. palustris* discriminately [35]. V-nitrogenase has not been reported in *R. capsulatus* and *R. rubrum* [36]. The genome of a PNSB *Rhodopila globiformis*, which has been isolated from an acidic warm sulfur spring, encodes V-nitrogenase in addition to Mo-nitrogenase [37]. Among phototrophic purple bacteria that possess nitrogenase, *R. globiformis* is an exceptional species in terms of possession of nitrogenase isozymes. Among 65 genomes of phototrophic purple bacteria that encode Mo-nitrogenase, operons for Fe- and V-nitrogenases were found in 14 and 6 genomes, respectively. Although operons for V-nitrogenase were found only in genomes encoding Fe-nitrogenase, the study has firstly reported the genome of phototrophic purple bacterium that encodes V-nitrogenase but does not encode Fe-nitrogenase.

## 3. Regulation of Nitrogenase and Glutamine Synthetase (GS) Activities

Since consumption of ATP and reducing equivalents to reduce nitrogen gas in nitrogen fixation imposes significant burden onto metabolisms in the presence of preferred nitrogen sources such as ammonia, and nitrogenase is rapidly destroyed by oxygen [38], diazotrophic bacteria have control mechanisms to suppress their activity when ammonia synthesis is not required in the presence of sufficient nitrogen sources or when environmental conditions are unsuitable for the catalysis of the reaction. Nitrogenase activity is controlled at the transcriptional level and post-translational level. PNSB regulate nitrogen fixation in response to ammonia, Mo, Fe, oxygen, and light [36]. As they can generate ATP required for the nitrogenase-catalyzed reactions by anoxygenic photosynthesis [39], light intensity is an important factor to control nitrogen fixation under light anaerobic conditions [9,40]. In PNSB strains that have either Fe- or V-nitrogenase or both Fe- and V-nitrogenases in addition to Mo-nitrogenase, the nitrogenase activities are regulated by concentrations of Mo, V, and Fe, which are needed for syntheses of FeMo cofactor for Mo-nitrogenase [41], FeV cofactor for V-nitrogenase [42], and FeFe cofactor for Fe-nitrogenase [43]. As shown in *R. palustris*, nitrogenase activity is diminished in the Mo-starved cells [44], whereas the Mo content in cells cultivated in N-free medium increases by seven times compared to cells cultivated in ammonium-added medium [45]. The V-nitrogenase-dependent diazotrophic strain of *R. palustris* fails to grow in medium containing an undetectable level of V below 0.1 ppb [32]. These studies indicate the necessity of Mo and V for establishing the active Mo- and V-nitrogenases in order to support diazotrophic growth. On the other hand, Fe-nitrogenase of *R. capsulatus* is repressed by Mo at concentrations above 1 ppb but not by V [28,46]. In *A. vinelandii*, *anfH* gene expression is repressed in N-free medium containing 1 µM Na_2_MoO_4_ or 1 µM V_2_O_5_ [47]. On the other hand, 100 µM Na_2_MoO_4_ does not repress Fe- and V-nitrogenases, and 100 µM VCl_3_ does not repress Fe-nitrogenase in *R. palustris*, in which fixed nitrogen availability may be the primary signal to control the synthesis of Fe- and V-nitrogenases [32]. Repression of the alternative nitrogenase activities by Mo and V does not seem to be universal among the diazotrophic bacteria.

GS was considered to be implicated in the mechanism of nitrogenase inhibition by ammonium in several strains including *Rhodobacter sphaeroides*, *R. capsulatus*, *R. rubrum*, and *R. palustris* [9]. *R. capsulatus* growing phototrophically in different inorganic nitrogen sources assimilates ammonia through the reactions of the glutamine synthetase/glutamate synthase (GS/GOGAT) pathway [48]. When GS is inhibited by methionine sulphoximine, reductive amination of pyruvate to L-alanine is the obligate way for ammonia incorporation. Alanine synthesis by the reductive amination of pyruvate is catalyzed by NADPH-dependent L-alanine dehydrogenase without consumption of ATP, followed by formation of pyruvate and L-glutamate catalyzed by L-alanine: 2-oxoglutarate aminotransferase. Hence, the GS/GOGAT pathway has priority in the assimilation of ammonia. As shown in *R. capsulatus* cultures, GS activity is inhibited by excess ammonium in the adenylylated (AMPylated) form [49]. However, it has been demonstrated that nitrogenase inhibition by ammonium and adenylylation of GS are independent processes in *R. palustris* [50] and *R. capsulatus* [9]. GS activity is also regulated at the transcriptional level. It has been demonstrated in *R. rubrum* that GlnB, which is a homolog of P_II_ proteins (small homotrimeric signal transduction proteins), plays a key role in transcriptional regulation of GS encoded in *glnA* via the NtrB–NtrC two-component regulatory system [51].

### 3.1. Transcriptional Regulation That Controls Syntheses of Nitrogenases

In γ-proteobacteria, the transcriptional control of *nif* genes is maintained by a regulatory complex comprising an enhancer-binding protein (NifA), which activates transcription at σ^54^-dependent promoters through ATP hydrolysis, and a sensor protein (NifL), which inhibits NifA activity in response to fixed nitrogen and external concentrations of molecular oxygen [52]. σ^54^ recognizes promoters with consensus sequences located at the positions −24 and −12 [53]. In *A. vinelandii*, *nifA*-like genes, designated *anfA* and *vnfA*, have been identified [54]. The synthesis of NifA is required for synthesis of Mo-nitrogenase but is not required for synthesis of either Fe- or V-nitrogenase. The gene products AnfA and VnfA are required for syntheses of Fe- and V-nitrogenases when *anf* and *vnf* genes are derepressed under Mo/V-deficient and Mo-deficient conditions, respectively. *A. vinelandii* and *Klebsiella pneumoniae* belong to γ-proteobacteria, and the NifL–NifA transcriptional regulatory system is thought to be common in this group [53]. On the other hand, most PNSB are classified as α-proteobacteria, and they seem to lack NifL [55]. A genome of purple sulfur bacterium *Halorhodospira halophila*, which is classified in γ-proteobacteria, also encodes *nifA*, in which the deduced amino acid sequence does not conserve the residue that interacts with NifL. Regulation of Mo- and Fe-nitrogenase activities by ammonium is well characterized in *R. capsulatus* [56]. *R. capsulatus* contains two functional copies of *nifA*, *nifA_1_*, and *nifA_2_* whose N-terminal 19 or 22 amino acid residues differ from each other [57]. In *R. capsulatus*, though NifA inhibition by NifL has not been reported, the NifA activity is known to be controlled post-translationally. GlnB and GlnK are responsible for the post-translational ammonium inhibition of NifA activity [58]. In enteric bacteria, global responses to changes in nitrogen status are mediated by the nitrogen regulation (ntr) system [59]. The ntr system is considered to be comprised of a P_II_ homolog GlnB, uridylyltransferase GlnD, and the NtrB-NtrC two-component regulatory system. GlnK is the second P_II_-like protein. In *glnB*-*glnK* double mutants of *R. capsulatus*, the inhibition of NifA activity by ammonium is completely abolished, while, under constitutive expression of *anfA* in Δ*nifHDK* background, AnfA activity is still suppressed by ammonium [58]. It is likely that, in the presence of ammonium, deletion of *glnB* and *glnK* genes results in derepression of *nifH* through the transcriptional activation by NifA, while it does not result in derepression of *anfH* probably due to no interaction of GlnB and GlnK with AnfA. On the other hand, although NifA is also required for the transcriptional activation of *nif* genes in *R. rubrum*, the function of GlnB on regulation of NifA activity is different from that in *R. capsulatus*. Neither a putative NtrC binding site nor a σ^54^-dependent promoter is found at the upstream region of *nifA*, and expression of *nifA* is constitutive in *R. rubrum* [60]. Therefore, the NifA activity is regulated not at the transcriptional level but at the post-transcriptional level. Uridylylated GlnB formed under catalysis of GlnD is required for NifA activation under ammonium-limiting conditions [60].

In the NtrB-NtrC two-component regulatory system of *R. capsulatus*, the enhancer-binding NtrC activates transcription of σ^54^-independent promoters [61], and the activation by NtrC is required for transcription of *nifA_1_*, *nifA_2_*, and *glnB* [62]. NtrC also activates transcription of *anfA*, which seems to be one of controlling points of Fe-nitrogenase activity in response to ammonia and Mo [56,63]. Thus, NtrC binding to the enhancer region is a crucial step, which is needed for the activation of *nif*- and *anf*-gene transcription by NifA and AnfA, respectively. Related to the transcriptional activation by NtrC, a mutation in *glnB* results in the constitutive expression of *nifA* and *anfA* in *R. capsulatus* [58]. In the proposed regulatory model, NtrC is deactivated by GlnB, which binds to NtrB and controls NtrB activity. Transcriptional regulations of nitrogenase and GS activities are summarized in Figure 1.

### 3.2. Reversible Inhibition (Switch-Off) of Nitrogenase Activity

Ammonium [66] and darkness [67] have been shown to cause a reversible inhibition (switch-off) of nitrogenase activity in PNSB. Ammonium inhibition has been also found in purple and green sulfur bacteria, as well [68]. In the reversible inhibition, a P_II_ homolog GlnB plays an important role on regulating activities of the enzymes responsible for post-translational covalent modification to nitrogenase [64]. Dinitrogenase reductase (Fe protein) is inactivated in response to stimulus via ADP-ribosylation by dinitrogenase reductase ADP-ribosyltransferase (DraT), and this modification can be reversed by dinitrogenase reductase-activating glycohydrolase (DraG) with opposing activity. Genes encoding DraT and DraG and their functions in *R. rubrum* have been reported [69,70]. A *draT* mutant cultured under nitrogenase-derepressing conditions does not show the switch-off with addition of ammonium or transition to darkness.

When cells of *R. capsulatus* are exposed to ammonium and darkness, both Fe proteins of Mo-nitrogenase and alternative Fe-nitrogenase are covalently modified [71]. In *R. palustris*, both V-nitrogenase and Fe-nitrogenase activities are inhibited, and Fe proteins of the alternative nitrogenases are post-translationally modified, when cells are exposed to ammonium [72]. The alternative nitrogenases are also substrates in ADP-ribosylation by DraT. A P_II_ homolog GlnB is required for the switch-off of nitrogenase activity via ADP-ribosylation of Fe protein catalyzed by DraT in *R. rubrum* [60,73] and *R. capsulatus* [58]. The GlnB* variants of *R. rubrum*, which are more readily uridylylated than wild-type GlnB, uniformly show poorer interaction with DraT, suggesting that unuridylylated GlnB might interact with DraT [74]. It has been demonstrated in *R. rubrum* that an activity of DraG in the cytosol is regulated with its association to and dissociation from the membrane via the association of DraG with a P_II_ homolog GlnJ [75]. In a *R. rubrum* mutant lacking the putative ammonium transporter AmtB_1_, which does not exhibit ADP-ribosylation and switch-off of the activity, DraG is mainly localized to the cytosol. GlnJ of *R rubrum* has higher affinity for an AmtB_1_-containing membrane than the other P_II_ homologs GlnB and GlnK [76]. This interaction strongly favors unuridylylated GlnJ and is regulated by levels of 2-ketoglutarate (2-KG) and the ratio of ATP to ADP. In *R. capsulatus*, a P_II_ homolog GlnK associates with AmtB in cellular membrane in response to an ammonium addition [77]. 

A DraT/DraG-independent switch-off mechanism that might block either the ATP supply or the electron supply to nitrogenase has been observed in *R. capsulatus* [78]. It seems likely that the second regulatory response is synchronous with ADP-ribosylation, and it is responsible for the bulk of the observed effects on nitrogenase activity. 

It has been demonstrated in *R. palustris* that nitrogenase activity in the *glnB* and *glnK1* mutants are susceptible to switch-off by ammonium, whereas nitrogenase activity in *glnK2* mutant is insensitive to ammonium addition [79]. The results suggest that GlnK2 is responsible for activation of DraT2, which is involved in ADP-ribosylation of Fe protein and, therefore, post-translational regulation of nitrogenase activity. GlnK2 is a P_II_ protein regulated by NtrC, and *glnK2* gene is upregulated in diazotrophically grown cells compared to cells grown with ammonium.

### 3.3. Post-Translational Regulation of GS

P_II_ homologs GlnK and GlnB are essential in nitrogen regulation due to this protein family’s ability to sense internal cellular ammonium levels and control cellular response. In *R. palustris*, GlnK1, GlnK2, and GlnB undergo uridylylation under ammonium-starved and nitrogen-fixing conditions [80], presumably to activate the AmtB ammonium transporter via loss of interaction and deadenylylation of GS via enhancement of adenylyl-removing activity in GlnE [64]. It has been shown in enteric bacteria that GS is highly responsive to cellular nitrogen status at post-translational modification via GlnE, which catalyzes covalent attachment and removal of an adenylyl moiety [81]. Adenylylation is stimulated under ammonium-sufficient conditions by unuridylylated P_II_, which enhances the adenylyl transferase activity of GlnE to inactivate GS. *R. rubrum* Δ*glnE* strains do not inactivate GS in response to an increase in ammonium concentration, indicating that GlnE is responsible for adenylylation of GS in *R. rubrum* [82]. *R. capsulatus* possesses two genes coding for ammonia transporters, *amtB* and *amtY*. Dysfunction of *amtB* but not *amtY* results in loss of the ADP-ribosylation in Fe protein and the switch-off of in vivo nitrogenase activity in response to ammonium addition [83]. Regulation of GS activity in response to ammonium addition is not affected in these mutants. AmtB seems to function as an ammonia sensor for the processes that regulate nitrogenase activity. Post-translational regulations of nitrogenase and GS activities are summarized in Figure 2.

### 3.4. Effects of Oxygen on Nitrogenase

Molecular oxygen causes irreversible damage to nitrogenases. Dinitrogenase reductases (Fe proteins) are more sensitive to inactivation by oxygen than dinitrogenases [84], because a [4Fe–4S] cluster of Fe protein that functions as the metallocenter of electron transfer is irreversibly destroyed by oxygen [85]. In addition to the irreversible damage, it has been shown that oxygen also causes a reversible inhibition of the enzyme activity in vivo in *R. sphaeroides* and *R. capsulatus* [86]. The reversible inhibition of the activity by oxygen is considered to be due to a diversion of electron transfer to the terminal electron acceptor (i.e., oxygen) from the substrates of nitrogenase [84]. This diversion of electron transfer is also caused within anaerobic respiration with sulfite in *R. sphaeroides* and *R. capsulatus* and with nitrite in a denitrifier *R. sphaeroides*. As shown in *R. capsulatus* exposed either suddenly or at a steady state to increased oxygen concentrations [10], ADP-ribosylation of Fe proteins does not seem to participate in the reversible inhibition by oxygen.

Exposure to oxygen irreversibly inactivates the Mo-, V-, and Fe-nitrogenases. To protect from the irreversible inactivation, *R. capsulatus* coordinately synthesizes Mo-nitrogenase and FdxD, the latter of which is encoded in an *fesII*-like gene *fdxD* [87]. FdxD is a [2Fe–2S] ferredoxin (Fd), which mediates conformational protection by interaction with Mo-nitrogenase. Expression of *fdxD* occurs under nitrogen-fixing conditions via transcriptional activation by NifA_1_ and NifA_2_, but not in the presence of ammonium. *R. capsulatus* potentially fixes nitrogen under aerobic conditions with light irradiation [88]. Hydrogen gas evolved by the nitrogenase reaction seems to act as an electron donor to the respiratory chain to participate in the scavenging of oxygen and the protection of nitrogenase from oxygen damage.

Nitrogenase-catalyzed reactions need low-potential electrons from Fd or flavodoxin (Fld) [89]. Fix complex and *Rhodobacter* nitrogen fixation (Rnf) complex generate reduced Fd from NADH/NADPH. In PNSB, *fixABCX* genes have been reported in *R. rubrum* [90], whereas *rnf* genes have been reported in *R. capsulatus* [91]. The distribution report of Fd- and Fld-reducing enzymes encoded in 359 genomes of putative diazotrophs has revealed that aerobic, facultative anaerobic, or anoxygenic phototrophic bacteria are largely dependent on Fix and, to a lesser extent, pyruvate-Fld oxidoreductase and Rnf [89]. The reduction of Fd or Fld that needs low-potential electrons in oxic environments is challenging, and aerobic, facultative anaerobic, and some anoxygenic phototrophic diazotrophs might acquire these functions to overcome the limitation in the delivery of electrons to nitrogenase. 

## 4. Contribution of Nitrogen Fixation by PNSB in Rice Cultures

Rice is one of the most important staple foods in the world, and the contribution of photosynthetic bacteria such as cyanobacteria and purple sulfur bacteria to nitrogen fixation in flooded rice cultures has been suggested [92]. Researchers have also paid attention to nitrogen fixation of PNSB in paddy fields and have examined application of PNSB to biofertilizers for rice cultivation [93,94,95]. Rice fields represent unique aqua–terrestrial ecosystems with tremendous diversity of soil microbes, soil fauna, and plants [96]. Though PNSB are regarded as aquatic organisms, it has been shown that the sediment samples from paddy fields are also a good source of PNSB when compared with the water samples [97]. Shallow paddy fields allow sunlight to penetrate to the sediment, organic matters provided by rice could be available carbon and energy sources for microorganisms including PNSB, and the oxidation-reduction potential in sediments of paddy field is a suitable range for promoting PNSB growth. These factors are of great advantage to PNSB inoculated to paddy fields in colonizing, growing, and functioning as biofertilizers in the environments.

### 4.1. Isolations of PNSB from Rice Fields

*Rhodopseudomonas* spp. are the most frequently reported species of PNSB isolated from rice paddy fields. Their abilities such as salt tolerance and production of 5-aminolevulinic acid (ALA) that reduces salt stress for plants [98,99], availability of organic carbon sources [100], availability of nitrogen and organic carbon sources [101], enhancement of rice growth and grain yield in acid sulfate soils with low available phosphorus [102], and plant growth promotion via enhancement of nitrate uptake and accumulation of plant hormone auxin [103] have been characterized. New species of the genus *Rhodopseudomonas* isolated from paddy soils include *Rhodopseudomonas telluris* sp. nov. [104], *Rhodopseudomonas pentothenatexigens* sp. nov., and *Rhodopseudomonas thermotolerans* sp. nov. [105]. Among 18 PNSB strains isolated from a paddy soil, 17 strains are *R. palustris*, and the other is *Rubrivivax gelatinosus* [106]. These strains exhibit enhancement of soil nitrogen fixation and mitigation of methane emissions in paddy soils. Thus, PNSB other than a genus *Rhodopseudomonas* have also been isolated. *Rhodovastum atsumiense* gen. nov., sp. nov., a phototrophic α-proteobacterium with nitrogen-fixing activity, has been isolated from paddy soil [107]. *R. sphaeroides* and *R. palustris* isolated from paddy fields have been treated with several pesticides to investigate effect of the pesticides on PNSB hydrogen metabolisms [108]. Among PNSB isolated from various paddy fields, *R. palustris* produces plant growth-promoting substances, ammonium, ALA, and indole-3-acetic acid (IAA); reduces methane emission from a mixture of soil slurry and ground rice straw; and removes Cd and Zn from the heavy metal-supplemented medium, whereas *R. gelatinosus* produces ammonium, ALA, and IAA and removes Cd and Zn [109]. *R. capsulatus* isolated from paddy soil produces carotenoids and IAA [110].

The monitoring of bacterial population has revealed that the number of PNSB in paddy fields increases gradually after transplanting, reaches its maximum at maximum tillering stage, and thereafter declines toward harvest time [111]. The metagenomic analysis by next-generation sequencing indicates that the PNSB population ratios in bacterial communities of Japanese paddy soils are not significantly influenced with the sampling months and fertilization conditions (Figure 3). The population ratios of PNSB suggest that the number of PNSB in the paddy fields might change in conjunction with total bacterial population if it is taken into account that total bacterial population in paddy fields increased after the tillering stage and reached the maximum value around the maturing stage [112]. Bacterial community analysis in metagenomes of five paddy soils collected from southwest China clarified that a PNSB genus *Rhodoplanes* was one of the most abundant genera with average relative abundances greater than 1% and was dominant in at least three rice paddy soils [113]. The relative abundances of PNSB reported in the Chinese paddy soils seem to be equivalent to those measured in Japanese paddy soils (Figure 3). As it is conceivable that the photic and microaerobic zones of the paddy field are habitats of PNSB such as *Rhodopseudomonas* spp., *Rhodobacter* spp., and other PNSB, physiological, metabolic, and genetic information could be useful in the application of PNSB to produce ammonia in a paddy field. 

### 4.2. Inoculation of Rice Cultures with PNSB Cells

Inoculation of rice cultures with PNSB cells in hydroponic, pot, and lysimeter experiments has been performed in order to examine effects of the inoculation on rice cultural parameters and nitrogen fixation by PNSB. Cells of *R. capsulatus* were inoculated to hydroponic cultures of four varieties of rice seedling with or without ammonium nitrate as a nitrogen source [115]. It becomes evident that the nitrogen gains and vegetative growth enhancement during the 3-week cultivation are primarily due to nitrogen fixation and that the inoculation gives benefits to rice seedling growth almost commensurate with the addition of ammonium nitrate. Pot and lysimeter experiments, in which *R. capsulatus* cells were inoculated to the roots of rice seedlings in combination with graded levels of ammonium sulfate, suggested that the inoculation could reduce 50% of the optimum nitrogen fertilizer quantity while keeping the grain yield and additionally increased the grain yield at the optimum nitrogen fertilizer quantity [116]. When effect of PNSB single strain supplementation on rice seedling growth and nitrogen content was investigated in hydroponic cultures containing ammonium chloride, the best growth (plant height, root length, and dry weight) and the highest nitrogen content after 30 days of sowing were observed in cell supplementation of *Rhodobacter* sp. isolated from paddy soil [117]. Inoculant cells of *R. palustris* significantly increased total nitrogen and ammonium concentrations in two different paddy soils [102]. Although nitrogenase activity in these soils was not measured, the inoculants composed of single strain and a mixture of four different strains increased ammonium concentration (mg/kg dry soil weight) from 20.6 without inoculation to 24.0 and 23.3 with Hon Dat soil and from 46.0 without inoculation to 50.8 and 52.2 with Phung Hiep soil, respectively.

Besides laboratory experiments, effects of PNSB inoculation on growth and grain yield of rice have been investigated in the field studies. *R. capsulatus* cells as a biofertilizer were inoculated to two different flooded paddy fields, and growth and yield of rice were compared with incorporation of chemical nitrogen fertilizer and farmyard manure [93]. Growth and grain yield were significantly higher in the PNSB-inoculated sites than in uninoculated sites in both fields. Though nitrogenase activity in soil was not measured, nitrogen content (%) in the grain significantly increased from 0.94 to 1.10 with the PNSB inoculation in comparison to observed increases from 0.89 to 1.12 with the incorporation of chemical nitrogen fertilizer and from 0.98 to 1.06 with the incorporation of farmyard manure. Application experiments using three strains of *R. palustris* in comparison with a commercial organic fertilizer (COF) were carried out in two different sites in Thailand during August to December and January to April [94]. The application frequency of all fertilizers was every 2 weeks during the vegetative stage and every week during the reproductive and maturation stages. The PNSB populations within inoculated plots increased in the first 40 days of plantation, was kept stably to day 80, and finally declined, compared to those in the plots without inoculation and the plots where COF was applied. The application of PNSB or COF also increased grain yields. 

### 4.3. Environmental Factors Affecting Nitrogen Fixation of PNSB

It has been demonstrated in surface soil slurries of rice fields amended with rice straw that acetylene-reducing nitrogenase activity by PNSB competes with methane emission by methanogens [118]. Light exposure that enhances nitrogenase activity in PNSB has a suppressive effect on methane emission, and inhibition of methane emission by a specific inhibitor, 2-bromoethanesulfonic acid, leads to increases in low-molecular-mass fatty acids and nitrogenase activities. This competition seems to create limitation of available substrates necessary for nitrogen fixation by PNSB and methane emission by methanogens. A negative correlation between acetylene-reducing nitrogenase activity or PNSB cell density and methane emission was shown in paddy soil slurries inoculated with *R. palustris* [106] and in paddy fields where three different *R. palustris* strains were separately inoculated [94].

A competition for substrates between sulfate reduction and nitrogen fixation was investigated in rice soil slurries mixed with rice straw in laboratory experiments [119]. When sodium sulfate at a concentration of 1 mg S/g dry weight soil was added to the soil slurries, light-dependent nitrogen fixation catalyzed by PNSB was repressed by sulfate-reducing bacteria. The inhibition of sulfate reduction and methane emission with molybdate prevented the competition for substrates, suggesting that stopping activities of competitive microorganisms such as sulfate reducing bacteria and methanogens is efficient for increasing nitrogen fixation in rice soil.

Although use of agricultural chemicals is beneficial to the improvement of crop productivity and reduction of farmers’ labor, it influences microbial community structures and microbial activities in crop fields. Effects of herbicide 2,4-dichlorophenoxyacetic acid, fungicides captan and carbendazim, and the insecticides quinalphos and monocrotophos on diazotrophic growth, nitrogenase activity, hydrogen photoproduction, and hydrogenase activity were investigated in *R. sphaeroides* and *R. palustris* [108,120]. These chemicals tended to inhibit diazotrophic growth, nitrogenase activity, and hydrogen photoproduction, while hydrogenase activities were less affected. For bioremediation of herbicide-contaminated soils, *R. capsulatus*, *R. sphaeroides*, *R. rubrum*, *Rhodopseudomonas acidophila*, *Rhodopseudomonas blastica*, *Rhodopseudomonas viridis*, and *Rhodomicrobium vannielii* were grown in the presence of herbicide butachlor [121]. Species in genera *Rhodobacter* and *Rhodospirillum* were less influenced by butachlor than those in genera *Rhodopseudomonas* and *Rhodomicrobium* in terms of nitrogen-fixing ability, indicating a difference in susceptibility toward the herbicide by genera or species of PNSB.

A *R. palustris* strain isolated from paddy soil synthesized CdS nanoparticles from Cd^2+^ and fixed nitrogen simultaneously [122]. Exposure to CdS nanoparticles and Cd^2+^ increased ammonium releases up to 2.83 and 1.11 times higher, respectively. The increase of ammonium release seemed to be due to upregulations of *nifH* and *vnfG*, which exhibited 2.57- and 1.60-fold changes by exposure to CdS nanoparticles. The strain could be beneficial to remediation of paddy fields that are contaminated with Cd and nitrogen fertilizer applications.

### 4.4. Application of PNSB Isolated from Harsh Environments to Biofertilizer

Acid-resistant *R. palustris* strains with biofertilizer and biocontrol properties have been isolated from peat swamp forests because approximately 50% of the world’s potential agricultural land is presumed to be acidic, and soil acidity has become a major obstacle to agriculture [123]. Among *R. palustris* strains isolated from peat swamp forests, a strain that is effective in providing high-affinity Fe (III) ligands siderophores, ammonium, and phosphate; strains that are effective in production of plant growth-promoting substances (siderophores, exopolymeric substances, IAA, and ALA); and a strain showing the highest inhibition against rice pathogens are potentially beneficial PNSB for rice cultivation in acidic soil.

*R. palustris* strains and a *Rhodopseudomonas harwoodiae* strain were isolated from acid sulfate soils and selected based on their resistance to acid and manganese [124]. High levels of Mn^2+^ in acid sulfate soils constitute a global health concern, and it is important to develop a suitable method of reducing Mn^2+^ toxicity in soil to ensure the safety of agricultural products including rice. These strains adsorbed Mn^2+^ more effectively with released exopolymeric substances and released ammonium by nitrogen fixation, phosphate by solubilizing phosphate from various phosphorus sources, siderophores, ALA, and IAA.

## 5. Future Directions

While the studies described above show effectiveness of PNSB inoculation on bacterial growth and nitrogen fixation in rice cultures, several studies suggest difficulties in colonization of inoculated PNSB, enhancement of nitrogenase activity, and ammonium release in rice fields. The pot experiment to test the effects of single and co-inoculation of *R. palustris* and *Bacillus subtilis* to paddy soils on soil-resident bacterial communities as well as grain yield and agronomic traits of the rice plant was conducted [125]. Microbial inoculations significantly improved the rice yields. However, the relative abundance of inoculants was not significantly increased, indicating the colonization failure of inoculants. When potential biofertilizers were selected from PNSB isolated from paddy fields, 90 strains were isolated from heavy metal-contaminated paddy fields, and 145 strains were isolated from saline paddy fields [109]. A total of the 235 strains were examined in terms of ammonium release in nitrogen-free medium containing sodium acetate under microaerobic light conditions. Out of the 235 isolates, 7 (3.0%) and 29 (12.3%) strains were classified as medium-level and low-level ammonium releasers, respectively, whereas ammonium release was not detected for the residual 199 strains (84.7%). The results indicate that even under the nitrogenase-derepressing conditions, ammonium is not secreted into medium in a majority of PNSB in paddy fields. Contrary to the results demonstrating the beneficial effects of PNSB inoculation on nitrogen gain by nitrogen fixation, plant growth, and grain yield in rice cultures, *R. palustris* inoculation did not increase nitrogenase activity in a pot experiment [126]. When cells of *R. palustris* were inoculated once or three times into floodwater with or without rice straw during rice pot cultivation, the inoculation increased the grain yield of rice. However, acetylene-reducing nitrogenase activity associated with soils or rice straw residues was not influenced by the inoculation. In this study, the authors concluded that the inoculation into floodwater was not effective for enhancing nitrogen fixation in paddy soils and that the beneficial effect of the inoculation on the grain yield was due to the other functions of *R. palustris*. These results suggest problems to be resolved in considering the use of PNSB as an alternative nitrogen fertilizer in the future.

### 5.1. Protection of Oxygen-Labile Nitrogenases from Oxygen by Respiration

Oxygenic phototrophs such as cyanobacteria produce oxygen during photosynthesis due to their ability to use water as an electron donor, whereas anoxygenic phototrophs such as PNSB do not produce oxygen during photosynthesis due to their inability to use water as an electron donor [127]. Hence, inhibition of nitrogenase activity by the endogenous oxygen evolution does not take place in PNSB during light-dependent nitrogen fixation. On the other hand, it is possible that nitrogen-fixing PNSB in their habitats such as paddy fields are exposed to dissolved oxygen under microaerobic conditions. Therefore, strengthening protection of nitrogenases against oxygen in PNSB might be effective to improve their nitrogen fixing ability in paddy fields.

In respiratory protection of oxygen-labile nitrogenase observed in *Azotobacter* species, consumption of oxygen at the surface of diazotrophic prokaryotes protects nitrogenase from inactivation by oxygen [128]. Hydrogen could be an electron donor of respiration for consumption of oxygen. An active hydrogen-oxidizing system recycles all of the hydrogen lost through nitrogenase-dependent hydrogen evolution. It has been demonstrated in the legume–*Rhizobium* symbiose that the hydrogen-oxidizing system in hydrogen uptake-positive *Rhizobium japonicum* bacteroids benefits the nitrogen-fixing process by providing the respiratory protection and generating the ATP needed to the nitrogenase reaction [129]. On the other hand, in a PNSB *R. capsulatus*, neither the significant protection of nitrogenase by hydroperoxidase or catalase nor the presence of respiratory protection could be detected under aerobic conditions [130].

### 5.2. Elucidation of Nitrogen Fixation by PNSB under Microaerobic Environments

At low oxygen concentrations observed in microbial habitats such as paddy soils, aerobic respiration might reduce the oxygen concentration in the vicinity of nitrogenases in diazotrophs including PNSB. Carbon sources decomposed from dead organic matters such as rice straw and hydrogen evolved under catalysis of nitrogenase and hydrogenase could be electron donors of respiration in various microorganisms. Low-molecular-weight organic acids and alcohols formed through microbial degradation of rice straw are suitable carbon sources for growth of PNSB [97]. When the effect of soil bacterium *Bacillus subtilis* inoculation on diazotrophic growth of *R. palustris* in the presence of air was examined, the significant bacterial growth and biofilm formation took place only in the co-culture of both bacteria (Figure 4) [131]. The result suggests that respiration by *B. subtilis* might cause a decrease of dissolve oxygen concentration in medium and contribute to the occurrence of nitrogenase activity in *R. palustris*. This study indicates the applicability of bacteria that have higher respiratory activities and grow under microaerobic conditions in paddy fields as microbial co-inoculants. In PNSB inoculation, such co-inoculants may be effective in preventing switch-off of nitrogenase activity and irreversible destruction of nitrogenase protein by dissolved oxygen.

Three nitrogenase isozymes differ in the amount of hydrogen produced per nitrogen consumed, resulting in different limiting stoichiometries for the overall reactions [132]. V- and Fe-nitrogenases are significantly less active in reducing nitrogen, and the ratio of hydrogen formed per nitrogen reduced is 3 and 7, respectively, whereas Mo-nitrogenase reduces 1 mol nitrogen to produce 1 mol hydrogen. V- and Fe-nitrogenases require 24 mol ATP and 12 mol electron and 40 mol ATP and 20 mol electron for reduction of 1 mol nitrogen, respectively, whereas Mo-nitrogenases requires 16 mol ATP and 8 mol electron for reduction of 1 mol nitrogen. Using *A. vinelandii* purified enzymes, specific activities and total electron flux in Mo- and Fe-nitrogenases were investigated at 1 atm nitrogen pressure [133]. Hydrogen-specific activity per total electron flux calculated from hydrogen- and ammonia-specific activities was 1.7-times higher in Fe-nitrogenase than in Mo-nitrogenase. This result indicates that Fe-nitrogenase supplies hydrogen gas more efficiently than Mo-nitrogenase does by diminishing ammonia specific activities.

As described above, several strains of *R. palustris* can synthesize V- and Fe-nitrogenases in addition to Mo-nitrogenase, and it has been demonstrated that Mo does not strictly repress Fe- and V-nitrogenase activities and V does not strictly repress Fe-nitrogenase activity. Hence, regardless of presence or absence of Mo and V, *R. palustris* V- and Fe-nitrogenases could catalyze hydrogen production more efficiently and play a role to balance the benefit and loss for nitrogen fixation in increasing the electron flux to proton. Hydrogen produced might activate respiration by hydrogen-oxidizing microorganisms including *R. palustris* [134] and decrease oxygen concentration in the vicinity of nitrogenases. On the other hand, allocation of electrons from the Mo-nitrogenase-catalyzed reaction to the alternative nitrogenase-catalyzed reactions diminishes the yield of ammonia production per electron or ATP. Chromosomal insertion mutants of *R. palustris* that had *vnfHDGK* or *anfHDGK* genes separately inserted into the chromosome under *pucBa* or *cisY* promoter were constructed [135]. In these mutants, *pucBa* promoter enables gene expression under low oxygen tension, while *cisY* promoter enables constitutive gene expression. A chromosomal *vnfHDGK* and *anfHDGK* deletion mutant, together with the chromosomal insertion mutants, might reveal effects of the efficient hydrogen production catalyzed by V-nitrogenase or Fe-nitrogenase on nitrogen fixation under microaerobic light conditions when *R. palustris* would be inoculated into paddy fields.

### 5.3. Genetic Engineering of PNSB to Modify Regulation of Nitrogen Fixation by Ammonium

Presence of mineral nitrogen in agricultural soil inhibits both nodule formation and nitrogenase activity in the legume–*Rhizobium* symbiose [138]. Nitrogen fertilization affects nodule formation, and the usually recommended rates of 40–60 kg N/ha suppress nitrogen fixation. When effects of the application of rice straw, farmyard manure, municipal biowaste compost, and municipal biowaste charcoal on soil nitrogen fixing activity of a model paddy microcosm were examined in comparison with urea fertilizer, the addition of rice straw with the highest C/N ratio among the organic wastes stimulated nitrogen fixation [139]. Nitrogen fixation in the soils was inhibited more significantly with the increased concentrations of added urea-N. Therefore, PNSB that are released from the switch-off of nitrogenase activity and repression of nitrogenase activity by ammonium might be useful as biofertilizers for continuous synthesis of ammonium in nitrogen-fertilized paddy fields.

It has been reported in *R. palustris* that strains harboring single point mutation in the transcriptional regulatory gene *nifA* produce nitrogenase-dependent hydrogen production in the presence of ammonium [140]. A stable mutant of *R. palustris* containing a 48-nucleotide deletion between *nifA* regions encoding GAF domain and AAA^+^ domain synthesizes NifA that adopts an active conformation in ammonium-grown cells [141] and produces hydrogen from ammonium-containing fermentation products [142]. When *nifA* gene was overexpressed in a mutant of *R. sphaeroides*, hydrogen production was detected in the presence of 14 mM ammonium [143]. A point mutation in *nifA* enabled hydrogen production in the presence of ammonium in a *R. capsulatus* mutant [144] and a *R. sphaeroides* strain harboring a vector expressing the *nifA* L62Q allele [145]. *R. palustris draT2* deletion or *glnK2* deletion in addition to the 48-nucleotide deletion in *nifA* improved ammonium tolerance and produced hydrogen under higher ammonium concentrations, compared to those observed with only the 48-nucleotide deletion in *nifA* [146]. It is possible that these genetically engineered PNSB strains harboring the mutations in *nifA*, *nifA* overexpression, *draT2* deletion, and *glnK2* deletion would continue to fix nitrogen in the presence of ammonium by avoiding switch-off of nitrogenase activity and repression of nitrogenase biosynthesis when nitrogen gas is available under microaerobic conditions.

Template-free disruption method mediated by CRISPR/Cas12a developed for genome editing and transcriptional regulation in *R. capsulatus* with editing efficiency up to 90% would accelerate genetic construction of PNSB strains [147]. Considering the possibility that alternative nitrogenases in PNSB might efficiently supply hydrogen, a substrate for respiration to reduce oxygen concentration under microaerobic conditions, mutations in *vnfA*, *anfA*, or both *vnfA* and *anfA* that have the coded proteins adopt their active conformations may be important for efficiently producing hydrogen in the presence of ammonium. Deletion of genes encoding AmtB, an ammonia sensor for nitrogen fixation, and GS, an enzyme for assimilation of ammonia and its liberation into the external medium [148], would be effective in developing PNSB biofertilizers, which have enhanced ability to release ammonia in nitrogen-fertilized paddy fields.

## 6. Conclusions

The previous studies reporting isolation of PNSB have shown that the rice field is a suitable habitat for *Rhodopseudomonas* spp., *Rhodobacter* spp., and the other bacterial species. PNSB population in paddy fields increases gradually after transplanting, reaches its maximum at maximum tillering stage, and declines toward harvest time. Low-molecular-weight organic acids and alcohols formed through the decomposition of plowed rice straw might be substrates for their growth and production of their metabolites. The studies related to inoculation of PNSB biofertilizers to paddy fields have revealed that they participate in nitrogen fertilization of cultivated soil, promotion of plant growth, and increase in rice grain yield. However, on the other hand, there are several reports that the application of PNSB biofertilizers does not contribute to an increase of nitrogen fixation in cultivated soil. A possible cause for this dysfunction might be repression of nitrogenase synthesis and switch-off of nitrogenase activity by ammonia and/or destruction of nitrogenase proteins and inhibition of nitrogenase activity by oxygen. Understanding of microbial community changes in paddy soil and co-inoculation of cooperative microorganism(s) may be helpful in avoiding such negative responses of inoculated PNSB biofertilizer. Development of genetically modified strains that have better protection against oxygen and are less susceptible to these repression and inhibition seems to be important for maintaining nitrogen fixation ability of PNSB biofertilizers inoculated in paddy fields. An advantage of using PNSB as nitrogen fertilizers is that the knowledge of transcriptional and post-translational regulations of nitrogenase activity and the constructed mutant strains with modified regulation (e.g., *nifA* mutation and overexpression, *draT2* and *glnK2* deletions) are available. Genetic modification on the alternative nitrogenase systems in terms of regulation in response to ammonium (e.g., modification in AnfA- and/or VnfA-mediated regulation, constitutive expression of *anf* genes and/or *vnf* genes) might be effective in efficient production of hydrogen, an electron donor for microbial respiration to decrease oxygen concentration in the vicinity of nitrogenases, in nitrogen-fertilized paddy fields. If PNSB fertilizers could maintain stable nitrogen fixation without being negatively affected by environmental factors of paddy fields, they could further contribute to reduction of chemical fertilizer consumption and establishment of sustainable agriculture.

## Figures and Tables

**Figure 1 microorganisms-10-00028-f001:**
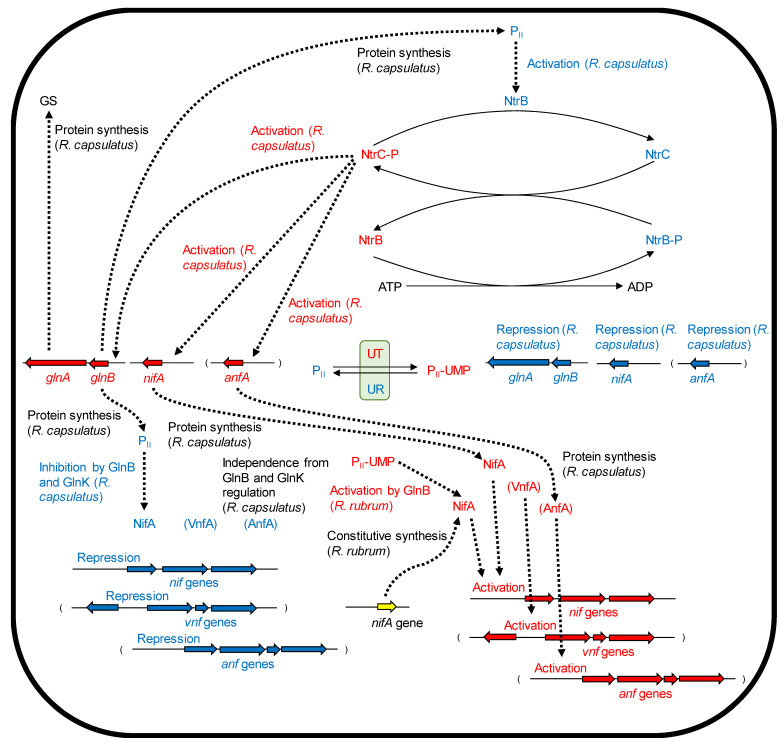
Transcriptional regulations of nitrogenase and GS activities in PNSB. Protein designations, gene maps and designations, and biochemical responses with red and blue colors mean their states under ammonia-deficient and ammonia-sufficient conditions, respectively. UR (uridyl-removing enzyme) and UT (uridyltransferase) activities are controlled by nitrogen status (2-oxoglutarate/glutamine) [64]. In a model shown in *Escherichia coli*, the NtrB autophosphorylation activity is inhibited by P_II_ [65].

**Figure 2 microorganisms-10-00028-f002:**
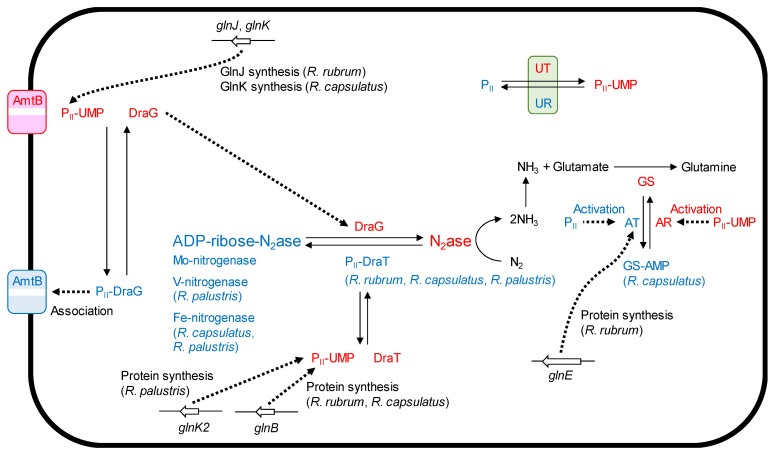
Post-translational regulations of nitrogenase and GS activities in PNSB. Protein designations and biochemical responses with red and blue colors mean their states under ammonia-deficient and ammonia-sufficient conditions or with light irradiation and darkness, respectively. UR (uridyl removing enzyme) and UT (uridyltransferase) activities are controlled by nitrogen status (2-oxoglutarate/glutamine) [64]. N_2_ase, nitrogenase; AT, adenylyl transferase; AR, adenylyl-removing enzyme.

**Figure 3 microorganisms-10-00028-f003:**
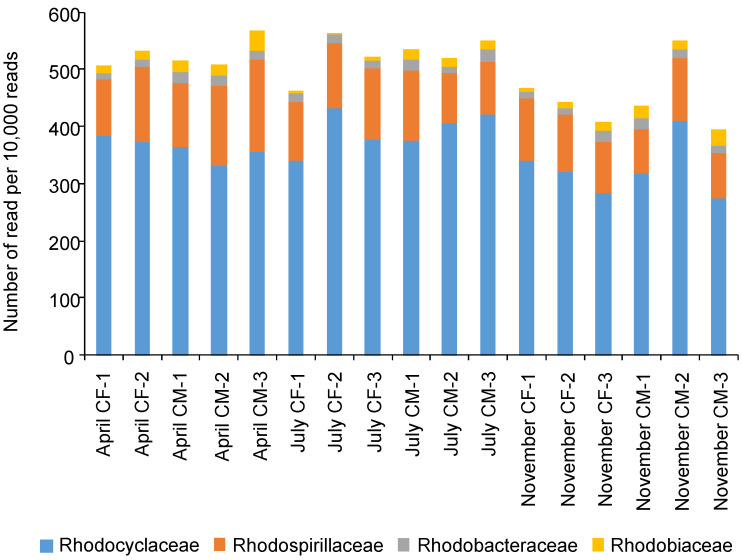
Number of 16S rRNA gene-sequence read identified as families to which PNSB belong in soil DNA collected from paddy fields. Soils were collected from the farm of the college of agriculture, Utsunomiya University, in April (plots before irrigation and plantation), July (plots during flooded culture with panicle initiation stage), and November (plots after harvest). CF and CM indicate chemical fertilizer and cow manure, respectively. DNA was extracted from soil using ISOIL for Beads Beating (Nippon gene, Tokyo, Japan). Metagenome sequencing was performed as described previously [114]. Sequence reads were processed and classified using Geneious Prime ver. 2019.2.3 (Tomy Digital Biology, Tokyo, Japan).

**Figure 4 microorganisms-10-00028-f004:**
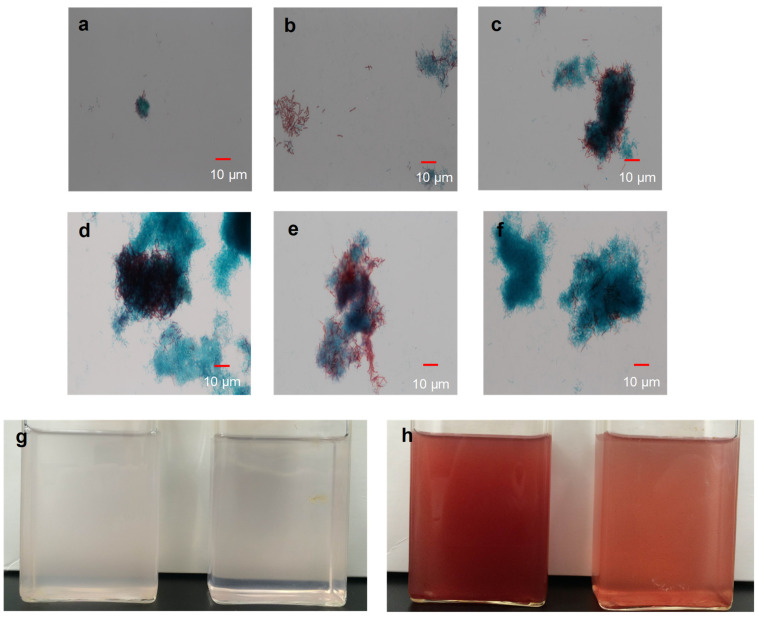
Diazotrophic growth and biofilm formation in the co-culture of *R. palustris* CGA009 and *B. subtilis* ATCC6633 in the presence of air. Microscopic observation of the co-culture after 1 day (**a**), 3 days (**b**), 4 days (**c**), 5 days (**d**), 6 days (**e**), and 7 days (**f**). *R. palustris* (blue–green) cells were stained with malachite green, and *B. subtilis* (red) cells were stained with safranin. After a versatile stain with malachite green [136], safranin replaced the malachite green in *B. subtilis* vegetative cells [137]. *R. p**alustris* pure culture (right) and the co-culture (left) after inoculation (**g**) and 7 days (**h**) are shown.

## Data Availability

The sequence data presented in this review are available on request from the corresponding author.

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
