# Peer review of "Potential of Phototrophic Purple Nonsulfur Bacteria to Fix Nitrogen in Rice Fields"

_microorganisms, 2021, doi:10.3390/microorganisms10010028_

Round 1
Reviewer 1 Report
I enjoyed this manuscript as the focus of the paper is new to me and the structure clear.
A few minor suggestions:
Could the author include a few more sentences and reference to describe how often and what is the abundance of PNSB (out of total microbe or bacteria community) typically seen in rice fields? if such data exist.
Figure 2 seem to be new data from the author, could the author provide more information about the data? Are the data obtained in a similar sampling and sequencing method that has been used before or described in a previous study of the author (provide reference)?
Line 326: "Conceivable."?
Figure 3: could the author also provide more information here so that the readers can understand the figures without referring to the original paper? For example, what is the function of the two dyes and what is the meaning of color difference.
Author Response
Thank you for your valuable comments. I totally agree with your opinions. I have carefully revised the manuscript according to the comments. The line numbers of revised manuscript in the response are shown below.

Reviewer 2 Report
This manuscript sums-up broad current knowledge relevant to nitrogen fixation by phototrophic purple nonsulfur bacteria and their potential use as nitrogen biofertilizer for rice cultures.
Comments to the author:
Lines 60-61: it should be “the gene nifH” and “the genes nifD and nifK”
Abbreviation of the mane of the microorganisms should be R. capsulatus, R. palustris, R. rubrum, …
Paragraph lines 105-132: more detailed information should be added on the GS activities.
Paragraph 77-87: remove last sentence. Move sentence to the next paragraph lines 89-103.
Paragraph 77-87: Add that R. palustris contains a Fe-nitrogenase.
Paragraph lines 89-103: “The genome of a purple nonsulfur bacterium Rhodopila globiformis, which has been isolated from an acidic warm sulfur spring, encodes two different types of nitrogenases, Mo- and V-nitrogenases but does not encode Fe-nitrogenase [34].” … “In the genome 98 survey study, 65 out of 80 phototrophic purple bacteria revealed the presence of Mo-nitrogenase, out of which 14 strains contained operons of Fe-nitrogenase and 6 strains contained operons of V-nitrogenase. Interestingly, alternative nitrogenases were found only in strains having Mo-nitrogenase. With the exception of Rpi. globiformis, V-nitrogenase was found in species which have both Mo- and Fe-nitrogenases.” This is confusing. Needs to be rephrased.
Figure 1: Figure 1 is difficult to read. Maybe Figure 1 should be slipped into several panels for each growth condition.
Figure 2: Add reference to Figure 2 legend.
Line 326-327: Fix the sentence.
Line 481: it should be PNSB.
References: names of microorganisms should be in italic. The name of the genes should be in italic.
The author is not discussing the role of electron transport system Fix or Rnf to the nitrogenase that are present in some PNSB.
Author Response

(The authors gave the same response as above.)
